# The CFHealthHub Learning Health System: Using Real-Time Adherence Data to Support a Community of Practice to Deliver Continuous Improvement in an Archetypal Long-Term Condition

**DOI:** 10.3390/healthcare11010020

**Published:** 2022-12-21

**Authors:** Robert D. Sandler, Martin J. Wildman

**Affiliations:** 1Sheffield Adult Cystic Fibrosis Centre, Northern General Hospital, Sheffield S5 7AU, UK; 2School of Health and Related Research (ScHARR), The University of Sheffield, Sheffield S1 4DA, UK

**Keywords:** adherence, behaviour change, cystic fibrosis, compliance, digital health interventions, habit formation, long-term conditions, quality improvement, self-care, trials within cohorts

## Abstract

CFHealthHub is a learning health system active in over 50% of adult CF Centres in England, supporting people with CF to develop habits of self-care around adherence to preventative inhaled therapy. This is achieved through the delivery of a behaviour change intervention, alongside collection of objective adherence data. As is common to long-term conditions, adherence to prescribed therapy is low, despite clear evidence of beneficial long-term impact on outcomes. This article explains how CFHealthHub is underpinned by coherent conceptual frameworks. We discuss how application of implementation and quality improvement strategies has facilitated CFHealthHub’s progression from a pilot study to a large, randomised control trial and now to a learning health system, becoming embedded within routine care. CFHealthHub is now able to support real-time health technology assessments, quality improvement and research trials and is in the process of being implemented in routine clinical care across participating centres.

## 1. Self-Care in Long-Term Conditions and Patient Activation

People with long-term conditions encompass 30% of the UK population, yet account for 70% of the total health care spend [1]. Providing patients with access to effective treatment improves outcomes. Unfortunately, up to 50% of patients do not take treatment as prescribed, and this low adherence reduces the benefits they may experience [2]. Interventions to improve adherence may use a range of strategies, but it is recognised that the choice of intervention is context-specific, depending on what is feasible and available for people with that long-term condition in that environment [3]. 

Patient activation, which describes knowledge, skills and confidence (often termed self-efficacy), is recognised as a determinant of how active a role patients are likely to play in their own care [4]. Within long-term conditions, higher levels of activation are positively correlated with adherence to treatment [5]. Patients with lower activation levels have higher usage of unscheduled emergency health care, which has important health economic implications [6]. Interestingly, there is only a moderate correlation between patient activation and socio-economic status, suggesting activation is not just a reflection of, or determined by, these factors [7]. 

Patient activation can be increased through targeted interventions, typically involving skill or knowledge acquisition and the development of confidence in ownership of their health. Patients with lower activation levels, who by definition are less interested in playing an active role in their own care, are less likely to take advantage of interventions on offer [8]. This is particularly interesting given the evidence that those with lower baseline activation tend to experience the biggest improvement in activation following an intervention [9,10]. This highlights how the way interventions to improve activation are designed and offered is an important health equality issue.

The Patient Activation Measure (PAM^®^) is a useful, quantifiable process measure, facilitating robust study of the impact of an intervention on activation [11]. The PAM^®^ is a validated questionnaire with 13 items. Responses to these items can be mapped to activation levels 1–4, representing a continuum. At level 1, patients are considered disengaged and overwhelmed, with the perspective that clinicians are in charge of their health. At level 4 they maintain healthy behaviours and feel they are their own health advocate [12]. Whilst an improvement in such process measures is useful and often more achievable in the constraints of a clinical trial, the real-world impact of behaviour change interventions on important health outcomes should remain a key priority for research. 

## 2. Learning Health Systems

A learning health system was originally described as “...one in which science, informatics, incentives and culture are aligned for continuous improvement and innovation, with best practices seamlessly embedded in the delivery process and new knowledge captured as an integral by-product of the delivery experience.” [13]. In 2012 and 2016, the Learning Health System Summits convened and refined the definition and published core values [14,15]. 

A recently refined learning health system description was of “a health system in which outcomes and experience are continually improved by applying science, informatics, incentives and culture to generate and use knowledge in the delivery of care” [16]. They are predicated on a continuous cycle with three recurring processes:1.Practice to Data

Data are captured from routine healthcare practice, for example an electronic health record with highly structured input fields

2.Data to Knowledge

Data is analysed to generate new knowledge, for example recognising patterns in health care utilisation within a hospital

3.Knowledge to Practice

The new knowledge is used for improvement initiatives, for example new protocols for staffing clinical areas based on identified patterns of health usage [16].

A workplace culture supportive of learning and improvement with the appropriate infrastructure to deliver this is essential in the development of a functional learning health system [17]. Co-production with key stakeholders, including patients, clinicians, and management informs development of learning health systems, which are designed to capture the metrics that matter and truly aid understanding of the current system performance. The coronavirus pandemic has enabled health systems to achieve accelerated digital transformation. Recognition that this progress must be maintained and adopted widely is reflected in recent health initiatives, such as “What Good Looks Like” within the UK National Health Service (NHS) [18].

## 3. Quality Improvement

Once an effective learning health system is in place, routinely reporting the metrics that matter, an ideal platform for Quality Improvement (QI) is created. QI involves the use of methods and tools to continuously improve quality of care and outcomes for patients [19]. The definition of “quality” is vague, though a definition adopted by the NHS in England asserts that quality of care is based on three criteria: safety; experience of care and effectiveness of care [20]. The Institute of Medicine broadens this to define quality in six dimensions: safe; effective; patient-centred; timely; efficient and equitable [21]. Taking these aspects into account, learning health systems can be designed to capture data to inform metrics within these six dimensions, to measure the impact of QI projects/interventions.

For QI to be successful, it is important not only to have access to the metrics that matter, but also to develop a culture in which front-line health care providers have the time and confidence to engage in these activities. Senior healthcare leaders can support this by ensuring there is a clear vision and objectives for the future, across all levels of an organisation, whilst supporting and enabling staff to participate [19].

## 4. CFHealthHub Learning Health System

Cystic fibrosis (CF) is an archetypal long-term condition, affecting approximately 11,000 people in the UK. People with CF have reduced life-expectancy of approximately 53 years, usually because of progressively deteriorating lung function on a background of recurrent or chronic chest infections, due to a genetic defect affecting production or function of the CF transmembrane regulator protein (CFTR) [22,23]. CF is the most prevalent inherited condition within Europe, and most commonly affects Caucasians [24]. As an inherited condition, the incidence is not dependent on typical social determinants of health, such as socio-economic deprivation, though of course these factors may contribute to outcomes and co-morbidity [25].

As with most long-term conditions, there are effective treatments in CF, including nebulised mucolytics and antibiotics which reduce both the risk of pulmonary exacerbations and the rate of lung function decline. Higher adherence to these treatments is associated with better outcomes [26]. Despite this, adherence is thought to be around 30–40% [27,28,29]. There are various methods of collecting information on adherence, including patient self-report, clinician report, therapeutic drug monitoring, medicines:possession ratio (, pickup/refill rates and electronic data capture [30]. In CF, self-report and medicines:possession ratio overestimate adherence, compared to electronic data capture [27,31]. If a person with CF has a deterioration in their lung function, it is important to determine if this could be explained by a new pathology necessitating a change in treatment or alternatively by suboptimal adherence to prescribed therapy. Access to accurate and reliable adherence data can help guide these clinical decisions.

CFHealthHub is a community of practice, active in 15 centres, representing over 50% of adult CF centres in England. CFHealthHub is underpinned by the clear objective: “to enable people with CF to live as normal a life as possible” by shifting care from rescue to prevention. Improvement has the best chance of occurring when a clear objective is shared and promoted within a health system [19]. One of the key aspects to achieving this objective is to promote adherence to preventative inhaled therapy. People with CF are provided nebulisers with electronic data capture capability when they enroll with CFHealthHub, allowing objective nebulisation usage to be recorded and adherence calculated against their prescribed regimen. Discussing detailed visual displays of objective nebuliser adherence data allows clinicians to share in a granular understanding of the lived experience of people with CF and come to a shared and respectful understanding of the challenges of treatment [32]. 

Further study in this area identified “habit” as a key influencer of adherence [33,34]. Subsequently, CFHealthHub was used to develop and deliver an intervention, aiming to help people with CF develop habits of nebuliser usage to increase adherence [35]. Complex interventions, such as those promoting behaviour change around adherence in CF, are often unsuccessful [36,37,38]. This may be due the absence of a theoretical or evidence-based approach, which is recommended for the development of successful complex interventions [39,40]. 

Within CFHealthHub, the Theoretical Domains Framework is used as the coherent conceptual model, which describes 14 key domains that may influence behaviour change and can be applied to a wide range of complex interventions [41,42]. These 14 domains can each be mapped to three distinct concepts of capability (C), opportunity (O) and motivation (M), all of which influence behaviour (B), and are represented in the COM-B model. Against this background, the CFHealthHub intervention was developed in accordance with the Behaviour Change Wheel, a framework supporting the design of behaviour change interventions [43]. The CFHealthHub platform and intervention were developed through co-production in a single centre before testing in a 3-centre pilot (2015) [35]. A mixed-methods process evaluation followed [44] and then evaluation in a 19-centre randomised controlled trial (RCT) (2017–2019) [45]. 

The CFHealthHub RCT (ACTiF) involved 607 participants and was the first, and currently only, RCT to demonstrate a sustained difference in adherence compared to a control arm in people with CF [45]. This supports the assertion that complex interventions grounded in theory, which are developed and delivered within a functional, cohesive learning health system are likely to be effective.

Since demonstrating the benefit of CFHealthHub in an RCT, the programme has moved to an implementation phase, with the intention to embed CFHealthHub in routine clinical care, providing objective adherence data and the habit formation intervention. The implementation strategy for CFHealthHub has been supported using microsystems improvement methodology developed in Dartmouth [46,47]. The implementation approach uses click analytics and “Plan, Do, Study, Act” (PDSA) cycles to integrate the complex intervention within the local context [48]. Each time members of the clinical team have an idea for change, we encourage them to use “The model for improvement” which consists of a series of PDSA cycles [49]. PDSA cycles are useful for implementing change and quickly assessing the impacts before repeating cycles. This allows clinicians to break down the complex process of making big changes within health systems into small, manageable, and measurable tasks. The intention is to refine each change through successive experimental PDSA cycles before standardising an effective process in the health system. Even once standardisation occurs, the process is not over, and should be revised as new technologies or challenges arise [50]. 

## 5. How CFHealthHub Works in Practice

Within the 15 UK adult CF centres, over 1500 PwCF are enrolled in CFHealthHub. CFHealthHub provides a platform for routine clinical care and research. The cohort multiple RCT design, now known as Trials within Cohorts, is proposed as a pragmatic, real-world extension to the traditional RCT to help answer important clinical research questions, including those about complex interventions [51]. It is based on the recruitment of large cohorts of patients with chronic conditions and the continuous collection of a range of clinical variables over time. 

The CFHealthHub learning health system, approved by London-Brent Research Ethics Committee (reference number: 17/LO/0032.) is an on-going cohort study to which people enrolling into CFHealthHub are invited to consent. The terms of this consent allow participants to contribute data as controls in future research. For example, should clinicians wish to perform a study within this cohort and compare a new intervention with “routine care”, participants who are not receiving the new intervention can form a control arm without the need for additional burden in terms of consent and data capture. Trials within Cohorts designed are best suited to trials with an open (unblinded) design with a “routine care” comparator arm and has the benefit of being able to include large numbers of patients relatively quickly and cheaply with minimal burden for those providing the comparator population. Using the Trials within Cohorts design can facilitate patient involvement in research, where control group participants simply carry on as normal and greater time and resource can be dedicated to recruiting more intervention participants who are difficult to reach. This is particularly important for complex behavioural interventions as engagement with clinical care may be on the causal pathway between the intervention and its effect on clinical outcomes.

Learning health systems provide a data-rich care environment that can potentially optimise several aspects of the complex systems that deliver long-term condition care and there are currently three major streams of work within CFHealthHub. Firstly, the national efficacy-effectiveness CFTR modulators optimisation programme (NEEMO) is a real-time health technology assessment nested within the CFHealthHub learning health system. It provides the setting of a prospective longitudinal cohort study using objective adherence and prescription data from CFHealthHub alongside data from the UK CF registry [52]. CF care was revolutionised in 2012 with the license of Ivacaftor, the first CFTR modulator, for which 8-10% of PwCF were eligible, based on their genotype. The key phase 3 trial demonstrated a 10.6 percentage point improvement in lung function for patients on Ivacaftor vs. placebo, but real-world data found only a 6 percentage point improvement [53,54]. Furthermore, after 5 years, lung function appears to return to the pre-Ivacaftor baseline [55,56]. With the high cost of CFTR modulators (approximately GBP 100,000 per patient per year), it is imperative to understand the reasons for this efficacy-effectiveness gap [57]. One explanation for this may relate to the reduction in use of preventative inhaled therapy, which has been seen since Ivacaftor was made available in the UK [58]. In the Ivacaftor RCT, the pre-trial treatment regimen (including preventative inhaled therapy) was maintained, and one consideration is that co-adherence of preventative inhaled therapy alongside CFTR modulators may impact on the overall clinical effectiveness, as it is thought to be the case in asthma, with respect to co-adherence of inhaled corticosteroids with the expensive biologic treatment, Mepolizumab [59]. 

In 2021, a combination CFTR modulator (Elexacaftor/Tezacaftor/Ivacaftor, marketed as Kaftrio in the UK), for which up to 80% of UK adults with CF may be eligible, was licensed. Like Ivacaftor, the key phase 3 of Kaftrio demonstrated a 14.3 percentage point improvement in lung function at 12 months [60]. Kaftrio is a similar drug to Ivacaftor in that it improves chloride channel function and like Ivacaftor may well be susceptible to pre-existing lung damage, necessitating on-going co-adherence to inhaled therapy. With the benefit of objective adherence data from CFHealthHub, NEEMO looks to explore the relationship between lung function, CFTR modulators and adherence to preventative inhaled therapy. 

The second workstream is the Easy Medicines for Burden Reduction and Care Enhancement (EMBRACE) project. EMBRACE aims to reducing medicines waste from excess supply, by ensuring “just-in-time” delivery by linking supply to actual use. Twelve centres recruited 275 people with CF and demonstrated that medicines:possession ratio over-estimates medication supply, compared to electronic data capture by around 15% [31]. A conservative estimate from this work was that GBP 822 per person/year could be saved if “just-in-time” medicines supply was driven by objective nebuliser usage data through CFHealthHub [31]. 

A key determinant of sustainability of digital platforms, such as CFHealthHub, is funding. Initially CFHealthHub was funded by a GBP 2.4 million National Institute for Health Research (NIHR) programme grant and then further by approximately GBP 8 million through an NHS England Commissioning for Quality and Innovation (CQUIN). Demonstration that CFHealthHub is not only beneficial for patients, through the ACTiF RCT, but also can generate cost savings, as shown in EMBRACE, is essential for adoption and spread. 

The UK National Institute for Health and Care Excellence (NICE) has recently approved the first Quality Indicators for CF, which form the basis of the third workstream. Quality is difficult to define and measure, hence NICE Quality Indicators “*measure outcomes that reflect the quality of care, or processes linked by evidence to improved outcomes*” and are used to identify where improvements are needed, set priorities for and support quality improvement, create local performance dashboards, benchmark performance against national data, demonstrate progress that local health systems are making on outcomes [61]. Within CF, median adherence to nebulised therapy has been selected as an appropriate process indicator to reflect the quality of CF care delivered at an individual centre [62,63]. As discussed, methods other than electronic data capture overestimate adherence, giving CFHealthHub the ideal opportunity to support delivery of the Quality Indicators. A key aspect of Quality Indicators is to provide centres insights into aspects of their service which could benefit from improvement. Within all conditions, the “quality” of care received at different centres will be variable, though this uncomfortable truth is rarely discussed openly. 

Identifying centres with higher median adherence allows a community of practice to work together to understand how this has been achieved and how other centres could learn and improve, an aspiration described by Atul Gawande in “The Bell Curve” [64]. CF in the UK is in a unique position in that there is a national registry in which all CF centres and 99% of people with CF are enrolled [52]. Therefore, the CF population denominator for each centre is known, and we can therefore meaningfully examine the reach of CFHealthHub at each UK centre, alongside the mean adherence for those who are enrolled. By providing insights into recruitment, distribution of data-logging nebulisers, engagement of those enrolled in CFHealthHub and ultimately accurate adherence data, we hope UK CF centres will be able to identify and set bespoke targets for improvement work.

## 6. Conclusions

CFHealthHub is a learning health system supporting a community of practice, caring for patients with an archetypal long-term condition, supported to pay attention to metrics that matter. A recent report from The Health Foundation cited CFHealthHub as the only disease-specific full learning health system with national reach in the UK [65]. The nature of CF as a condition, with a national registry and a small number of dedicated health care settings, makes total population denominators visible and can highlight inequity. This provides a unique opportunity to learn how we can use a data-rich learning health system to improve care, address inequalities and undertake research to address the priorities of the CF community. Similar principles could be applied to learning health systems in other long-term conditions, which could have more profound population-level health and economic benefits.

## Data Availability

Not applicable.

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
