# Peer review of "The CFHealthHub Learning Health System: Using Real-Time Adherence Data to Support a Community of Practice to Deliver Continuous Improvement in an Archetypal Long-Term Condition"

_healthcare, 2022, doi:10.3390/healthcare11010020_

Round 1

Reviewer 1 Report

Line 2: Please spell cystic fibrosis for the first reference before abbreviation of CF

Line 13-17: please divide into 2 sentences. As it reads now, it is too long to describe the intentions of the study or review.

Lines 32-34: The description of patient activation as a self-efficacy term is an interesting concept.

Line 46-47: should ‘is’ be ‘as’? Please reword the sentence.

Line 73-81: Section 1.3. QI: good, kudos. QI and data collection are always labor-intensive items with front-line staff members. Educating them on the importance of the results contributes to ownership of the process.

Line 82: you have referenced healthhub as one word in previous sentences. Should it be one word here?

Line 84-87: long sentence, please split.

Lines 105-122: good.

Line 152: good statement of critical thinking!!!!!

Line 178: You state there are three major streams of work, and include the first one. Did I miss the second and third?

Reviewer 2 Report

The CFHealthHub learning health system: using real time ad- herence data to support a community of practice to deliver con- tinuous improvement in an archetypal long-term condition.

It is a review that contributes to science. CFHealthHub is now able to support real-time health technology 17 assessments, quality improvement and research trials.

Page 2-line 48-More detailed and descriptive information can be given about The Patient Activation Measure.

Page 2-line 54-More detailed and descriptive information can be given about the learning health system.

Page 2-line 67-More detailed and descriptive information can be given about the Quality Improvement.

Page 5-line 248- The conclusion section could be written in a little more detail. Suggestion sentences can be increased.

Reviewer 3 Report

Dear Authors!

Your review is devoted to a comprehensive description of the possibilities of the CFHealthHub. To present the advantages of the CFHealthHub learning health system, the authors chose a right nosological form - cystic fibrosis. The authors characterize in detail this archetypal LTC, concisely and clearly describing the cohort of patients, the state of their registration in the UK, the main methods of therapy, including mucolytic, antibiotic and CFTR modulators. The adherence of the CF patients to nebulised mucolytics and antibiotics is the one of the most pressing questions for clinicians. It was this therapy that was chosen by the authors to describe the operation of the CFHealthHub in practice. Having given data on the costs of developing the CFHealthHub learning health system, high-cost modulatory drugs, the authors showed what savings can be achieved by increasing patient adherence to the inhalation therapy. The authors show the directions and opportunities for the development of the CFHealthHub and the role of all key stakeholders in this process. One can only thank the authors for such a clear statement of the need and role of the learning health system. One small note. Please add an explanation of the RCT abbreviation in the text so as not to refer readers to the reference title, containing the phrase “a randomised controlled trial.”
